# Par3DNet: Using 3DCNNs for Object Recognition on Tridimensional Partial Views

**Francisco Gomez-Donoso** *,† **, Felix Escalona** † **and Miguel Cazorla** †

University Institute for Computer Research, University of Alicante, PO BOX 99 Alicante, Spain;
felix.escalona@ua.es (F.E.); miguel.cazorla@ua.es (M.C.)

**\*** Correspondence: fgomez@ua.es

**†** These authors contributed equally to this work.

**Abstract:** Deep learning-based methods have proven to be the best performers when it comes to object recognition cues both in images and tridimensional data. Nonetheless, when it comes to 3D object recognition, the authors tend to convert the 3D data to images and then perform their classification. However, despite its accuracy, this approach has some issues. In this work, we present a deep learning pipeline for object recognition that takes a point cloud as input and provides the classification probabilities as output. Our proposal is trained on synthetic CAD objects and is able to perform accurately when fed with real data provided by commercial sensors. Unlike most approaches, our method is specifically trained to work on partial views of the objects rather than on a full representation, which is not the representation of the objects as captured by commercial sensors. We trained our proposal with the ModelNet10 dataset and achieved a 78.39% accuracy. We also tested it by adding noise to the dataset and against a number of datasets and real data with high success.

**Keywords:** 3D object recognition; point cloud object recognition; 3d-based deep learning

## 1. Introduction

In the last few years, with the advent of cheap 3D sensors such as Microsoft Kinect or Asus Xtion, object recognition in tridimensional data has been a blooming field with several successful works [1]. The problem that these approaches address is to recognize which object is represented in the input tridimensional data. This is important because, despite the current image-based object recognition systems perform very accurately, there are common situations where these systems are still prone to fail. For instance, image-based object recognition systems would fail wherever exist depictions of objects that actually are other objects, such as posters in the street or billboards depicting persons. This is because these systems rely its capabilities on recognizing certain visual features, but they can be easily fooled [2–4]. Furthermore, there are several studies that state the difficulties that humans have to distinguish between some tridimensional objects using only 2D data, when they are seen from unusual points of view [5]. Since neural networks operate according to principles similar to those of the human mind, it is not surprising that they suffer from the same limitations.

In addition, most tridimensional-based approaches in the state of the art are challenge-driven. Namely, the authors usually impose restrictions on their methods that are acceptable for the challenges they apply to, but that renders the approach useless for real life applications [6]. The most severe of these restrictions is that the methods usually take as an input a full tridimensional representation of the object. Nonetheless, this case would not occur when capturing 3D data with an actual sensor in real life because it is only being captured the part of the object that is facing the sensor.

Thus, in this paper, we propose Par3DNet: a deep learning-based architecture for 3D object recognition that takes a partial 3D view of the object as an input and is able to accurately classify it. As

there is a lack of large-scale datasets of partial views of real objects, we also focused on filling the gap between synthetic and real data. Despite the fact that we trained our system on synthetic data, it is able to generalize and perform accurately on real life objects.

Specifically, the main contributions of this work are:

- A deep learning-based architecture that is able to take 3D data as input and perform classification.
- The architecture is able to classify partial 3D data like the one provided by a Kinect sensor.
- The architecture is trained on synthetic data and performs accurately when fed real 3D data.

The rest of the paper is organized as follows: first, we review some relevant related works in Section 2. Then, we describe our approach and the data preprocessing method in Section 3. In Section 4, we show the results of the experiments we carried out to validate our proposal. Finally, we draw the conclusions and state the limitations and future works in Section 5.

## 2. Related Works

With the advent of low cost 3D sensors, researchers are focusing on deep learning methods for 3D object recognition. Following the classification presented in [7], we divide the methods in terms of the most used data representation.

**Raw pointcloud**. 3D data is represented as a unordered set of points within 3D space. They extract spatial features directly using nearest-neighbors and radius search. *PointNet++* [8] calculates features over the points and apply transformations to the data. Finally, it builds a global feature vector, that feeds a neural network for classification or segmentation purposes. This method learns hierarchical features from the points, introducing layers of sampling and grouping. *Escape from Cells* [9] represents the 3D point cloud as a balanced kd-tree of fixed size. Then, it calculates the vector representation of the points applying transformations from the leaf nodes to the root. *VoteNet* [10] proposes a novel technique based on Hough voting using a backbone network, implemented with *PointNet++* layers, that generates a subset of interesting seed points with their corresponding deep features. Each selected point votes for its corresponding object class and then these votes are grouped into clusters and processed to generate 3D bounding boxes with the final classification. *SplatNet* [11] takes raw pointclouds as input and extends the concept of 2D image SPLAT to 3D, allowing an efficient specification of filter neighborhood with the use of hash tables (easy mapping of 2D points into 3D space), and apply several bilateral convolutions to extract the features. *SO-Net* [12] proposes a method based on a self-organization mechanism to ensure point permutation invariance. This method models the spatial distribution of pointclouds by building a Self-Organizing Map (SOM), and then extracts hierarchical features from every point using their neighborhood. Finally, it generates a feature vector to describe the cloud globally. *Point-Voxel CNN (PVCNN)* [13] combines the sparse pointcloud representation with the performance of voxel-based convolutions to reduce sparse data access and improve the locality of the method. The authors introduce a novel hardware-efficient primitive, Point-Voxel Convolution (PVConv), which transforms the points into low-resolution voxel grids, makes aggregation of neighbor points with voxel-convolutions and convert them back to points via devoxelization. During the process, they apply point-based feature transformation to obtain finer-grained features.

**2D projections**. Input data is represented as 2D projections from different points of view of the 3D data, *slices*. These are the most common approaches, that use a single or multiple views of the object to feed a *Convolutional Neural Network*, as presented in [14]. Some works have focused on grouping views and training a boosting classifier to improve their performances [15]. Another approaches are based on the manual selection of the best views of the object to make the inference, as exposed in [16], that uses 3 orthogonal views that feed 3 independent neural networks. This group of techniques has the best classification performances, as exposed in ModelNet benchmark [17], but they need a full reconstruction of the object in order to generate multiple views, so they are not so reliable in real-world applications when dealing with occlusions and partial views.

**Voxelization**. Input data is represented as a discretization of the space around the data as an approximation of the original form. Every voxel usually contains a 0 or 1 indicating the presence or not of points inside. The original proposal from the creators of ModelNet, *3D ShapeNets* [18], represents the data as a cubic voxel and apply 3D convolutions with restrictions to obtain the vector representation. *VoxNet* [19] applies a 3D *Convolutional Neural Network* to this volumetric representation for classification purposes. In the case of *PointGrid* [20], it incorporates a constant number of points within each grid cell, using a technique they call point quantization, and stacks the points's coordinates as features for each cell, allowing the network to learn better representations of the local geometry of point clouds. However, not only have *CNNs* been used, but so have novel deep learning architectures: *Vconv-dae* [21] employs a convolutional denoising auto-encoder as a feature learning network, ref. [22] uses a *Variational auto-encoder* and [23] a *Generative Adversarial Network*. Similar to multiview 2D projections, researches such as *MO-VCNN* [24] explore the advantages of generating different rotations of the 3D models, bu changing azimuth and elevation angles, extracting high level features from every orientation and combining them into a final feature vector. Other works, such as *O-CNN* [25] and *OGN* [26] try to take advantage of the speedup of 3d convolutions obtained by using octree representations of 3D data, along with the possibilities of lower memory consumption compared with fixed size grid representations. The main issue of these techniques is the lack of precision when discretizing an object, which can lose fine details, and the enormous space requirements of the 3D convolutions in memory.

**Datasets**. There are some state of the art datasets that have captured single objects in isolation with real sensors for 3D object recognition, as presented in [27]. *RGBD Object Dataset* [28] and *MV-RED* [29] offer captures of 300 and 500 objects using Kinect v1, but without information of the pose. *BigBIRD dataset* [30] and *YCB Object and Model Set* [31] additionally incorporate pose information of 100 objects using an Asus Xtion Pro sensor. However, the biggest effort has been performed by *A large dataset of object scans* [32] that provides captures of more than 10000 objects using a PrimeSense sensor, with the camera pose computed directly from the data. There also are some other widely used datasets such as SUNRGBD [33], SCANNET [34] and S3DIS [35]. Nonetheless, these datasets are intended for object detection and segmentation, so it is unclear how to adapt them to enable a fair testing protocol for benchmarking object classification approaches.

The main problem with the real data is the difficulty to acquire ground truth for 3D pose estimation, that cannot be obtained without external hardware. To solve this issue, many researchers have adopted synthetic datasets. These datasets allow users to control several aspects more carefully, such as point density, pose or noise level. ModelNet [18] provides two different datasets *ModelNet10* and *ModelNet40* with thousands of CAD models for 10 and 40 different classes respectively. They are considered de facto standard datasets for 3D object recognition tasks. On the other hand, *ObjectNet* [36], with 100 categories and more than 40 thousand aligned 3D shapes, represents another of the biggest datasets for this purpose.

In view of existing methods, we propose an architecture focused on classifying 3D objects by just using one partial view, as the ones provided with sensors like Kinect, in order to improve the results in real applications. Most datasets offer complete views of the 3D objects, so we define a methodology to extract partials views from them, and therefore we take advantage of this large synthetic datasets to classify real 3D scans.

Thus, in this paper, we propose Par3DNet: a deep learning-based architecture for 3D object recognition that takes a partial 3D view of the object as an input and is able to accurately classify it. As there is a lack of large-scale datasets of partial views of real objects, we also focused on filling the gap between synthetic and real data. Despite we trained our system on synthetic data, it is able to generalize and perform accurately on real life objects.

## 3. Proposal

In this section we describe the Par3DNet architecture and the method to generate the input binary voxel representation. W also explained the preprocessing we applied to the CAD models to generate point clouds that depict partial views of the objects to train the architecture.

### 3.1. Par3DNet Architecture

The Par3DNet architecture is based on the PointNet [37]. It takes as an input a 3D volume, which represents the binary voxel representation of a point cloud. Then, this data is forwarded to a pipeline of convolutional layers with tridimensional filters, and dropout layers. Finally, a fully connected layer is in charge of performing classification. Figure 1 shows the architecture of Par3DNet.

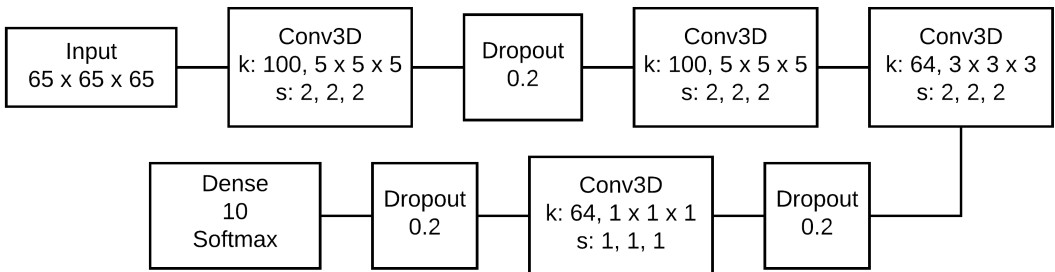

**Figure 1.** Architecture of Par3DNet. The network receives a binary voxel representation of the point cloud of shape $65 \times 65 \times 65$ as input. Then, it applies 2 consecutive 3D convolutions with 100 kernels of shape $5 \times 5 \times 5$ with an stride of 2 for every dimension. Later, it performs a 3D convolution with 64 kernels of shape $3 \times 3 \times 3$ with an stride of 2 for every dimension. Afterwards, it makes a $1 \times 1 \times 1$ kernel convolution to reduce the dimensionality of the previous layers. Finally, it calculates classification probabilities for the given classes of the dataset with a softmax function. Dropout layers inhibit a ratio of connections in each step and benefit the generalization capabilities of the network.

The binary voxel representation, which is the input layer, consists of a tridimensional volume of $65 \times 65 \times 65$ voxels. It is built by enclosing the 3D sample, which should be a point cloud, in a voxel grid. Each voxel is filled with a 1 if there are points in the space delimited by that voxel, or a 0 otherwise. This way, we generate a binary occupancy grid for each sample, which is then forwarded to the first 3D convolutional layer.

The voxel representation length, width and depth units are set to fit the metric size of each dimension of the input point cloud, so the voxel size is not fixed to a specific metric unit. This way, the absolute size of the objects is missing. Nonetheless, this is a desirable feature as the CAD models do not usually match the actual size of the objects they represent. In fact, the aspect ratio of the objects is actually not desired.

The specific size of $65 \times 65 \times 65$ voxels was chosen to accommodate at least 10 samples of each category in a batch. This is done to benefit a smooth learning and fast convergence at training stage. The size of the representation could be enlarged in order to capture finer topological features at the expense of higher memory requirements.

This unit is represented as the Input layer in Figure 1, followed by the size. Some samples of random point clouds converted to the input voxel representation can be seen in Figure 2.

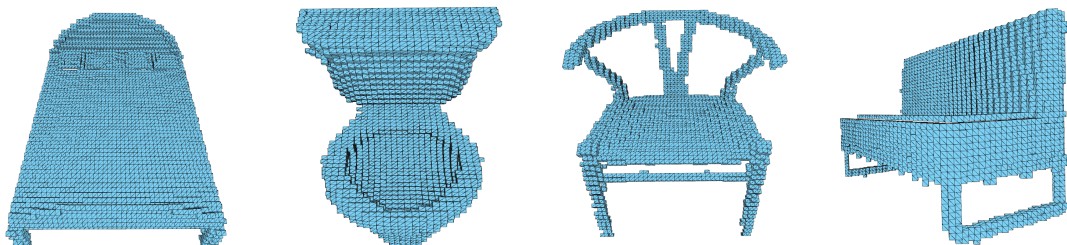

**Figure 2.** Some samples of random point clouds converted to the input voxel representation. Note that the representation does not cover the full object, but only the part that would be facing the sensor.

The tridimensional filters of the convolutional layers focus on learning the topological features of the samples, rather than the visual features which are learn by the 2D filters that are used in image data. In the Figure 1, the 3D convolutional layers are labeled as Conv3D. The number of filters of each layer are specified in the parameter $k$, followed by its size. The stride parameter is $s$. The activation function of every 3D convolutional layers is ReLU.

The dropout layers randomly inhibit a ratio of connections in each feedforward step. This is done to benefit the generalization capabilities, as disabling connections coerces the network to deal with missing features that are even different each time. Thus, these layers also enhance the performance when the architecture is fed with occluded objects that yield missing parts. They are represented as Dropout in Figure 1 alongside the ratio of disconnections.

Finally, the classification layer is a fully connected layer with 10 neurons. This parameter is set to fit the number of classes considered in out experiments. The activation function in this layer is softmax. In the Figure 1 this layer is labeled as Dense, followed by the number of neurons and the activation function.

### 3.2. Training Data and Preprocessing

As there is a lack of large-scale datasets of real-life objects with known poses, we adopted ModelNet10 [18]. This dataset is composed of 4900 3D CAD models of objects. They are categorized in 10 different classes: toilet, table, desk, sofa, night stand, monitor, dresser, chair, bed and bathtub. The models are manually aligned so all of them share the same pose. The authors of the dataset provide both training and test splits too. This dataset was chosen because the amount of samples is enough to train our architecture without requiring too much time. The Figure 3 shows random samples of ModelNet10.

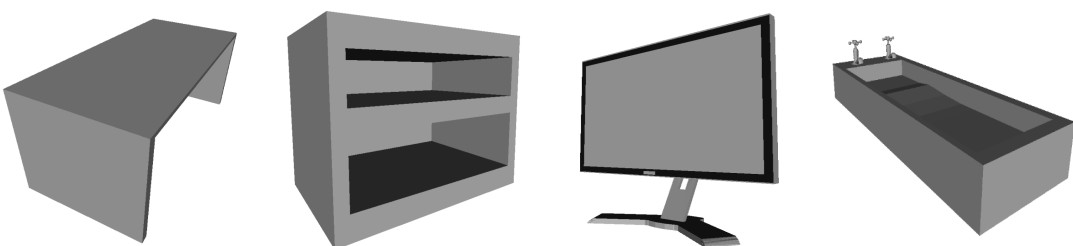

**Figure 3.** Some samples of the CAD models that compose the ModelNet10 dataset.

As mentioned before, our approach is able to classify point clouds that depict a range of objects provided only by a partial view of it, similarly as the data provided by commercial sensors. In this way, the point clouds do not represent the complete object, but the part of it that is facing the sensor, as depicted in Figure 4. As expected, the inner artifacts and the back of the object are missing in the point clouds.

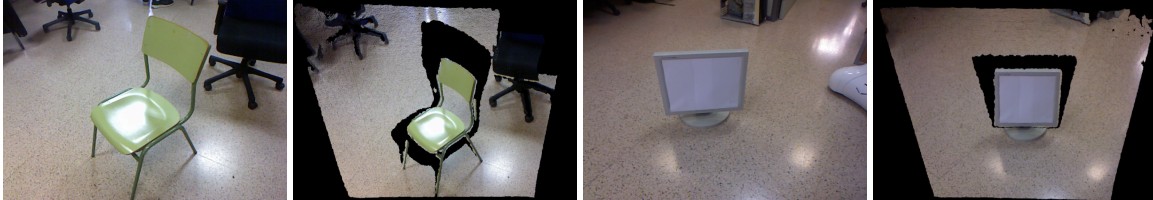

**Figure 4.** Samples of color images and the corresponding point clouds as provided by a Kinect. Note that the point clouds only depict the part of the object that is facing the camera.

In order to use the ModelNet10 dataset, which are CAD models, to our purpose, we first need to extract the point clouds of the objects as if they were captured by an actual 3D sensor. To do that, we located the CAD objects in the center of a tessellated sphere. Each corner of that tessellated sphere yields a virtual 3D camera which captures the tridimensional data of the object from different points of view. This way, we get 42 different views of each CAD object. It is worth noting that the views are, in fact, point clouds. This process is shown in Figure 5.

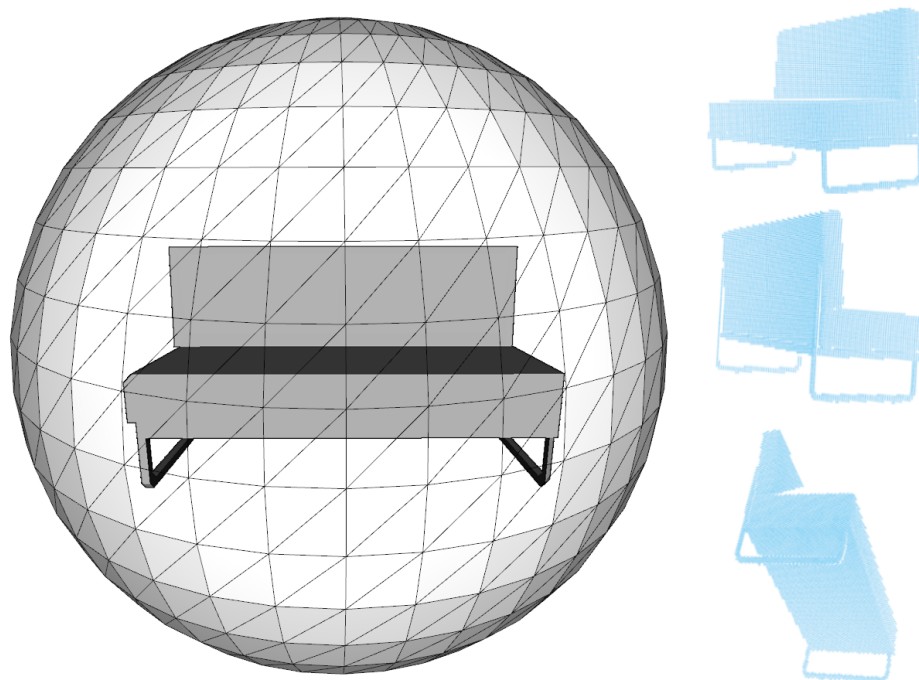

**Figure 5.** In order to get the samples, we located each CAD object inside a tessellated sphere (**left**). Each corner of the tessellated sphere has a virtual 3D camera that provides a partial, point cloud view of the object (**right**).

As a result, there are some views, namely point clouds, that are not interesting because it is unlikely to find that view in the wild. For instance, the point clouds that depict the underneath of the objects are discarded, and not considered for training nor for testing. Thus, we only consider the remaining 25 most relevant views. Finally, we got a dataset comprised of more than 76,000 training samples and about 18,000 testing samples. The distribution of samples per category can be seen in Figure 6. It is worth noting that, at this point, a sample is actually a point cloud that depicts a partial view of a ModelNet10 CAD object.

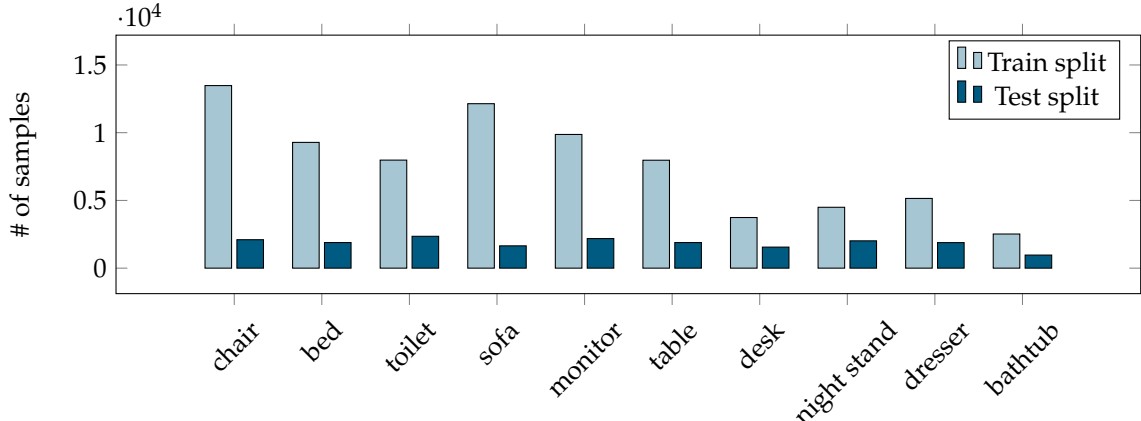

**Figure 6.** Number of samples in the dataset.

## 4. Experimentation

Once we prepared the data, we started the training process of the architecture. The *batch size* was 100 and the samples were randomly shuffled in each iteration. The *loss function* was *categorical crossentropy*, which was optimized with *Adam*. The starting learning rate was 0.0001. The architecture was trained for 2818 epochs, which took about 25 days in the hardware setup described below. We focused on the loss and accuracy values of both training and test splits to apply an early stopping criteria. Finally, the last epoch achieved a training loss of 0.021 and an accuracy of 99%, whilst the test loss was 1.471. We used the resultant model in all the experiments of this section.

The following hardware was used for all the experiments: an Intel Core i5-3570 with 16 GiB of Kingston HyperX 1600 MHz and CL10 DDR3 RAM on an Asus P8H77-M PRO motherboard (Intel H77 chipset). The system also included an Nvidia GTX1080Ti. The framework of choice was Keras 1.2.0 with Tensor Flow 1.4 as the backbone, running on Ubuntu 16.04. CUDA 9.0 and cuDNN v7.1 were used to accelerate the computations.

### 4.1. Results on the ModelNet10 Dataset

We fed the test samples to the architecture and we got the results shown in Table 1. The global accuracy in this case is 78.39%. As shown in the figure, the bathtub class only got 59% accuracy. This is understandable as this is by far the category with the fewest number of samples in the dataset. This fact is, thus, clearly reflected in the results. The desk and table categories yield bidirectional confusions. This is also expected as both categories depict barely the same topological data. In this case, the desk category achieves 41% accuracy, but the 23% of the samples were labeled as desk. The desk category also has a low number of samples compared with the table one, so this also explains the confusion. There is also a slight confusion between the nightstand and dresser categories for the same reason as that for the desk and table categories. The best performer is the chair category. This is due to the chair category being the one with the highest number of samples in the dataset.

**Table 1.** Confusion matrix of the ModelNet10 test split achieved by our model.

| | | Predicted | | | | | | | | | |
|---|---|---|---|---|---|---|---|---|---|---|---|
| | | Bathtub | Bed | Chair | Desk | Dresser | Monitor | Nightstand | Sofa | Table | Toilet |
| | bathtub | 59 | 5 | 3 | 2 | 3 | 4 | 1 | 15 | 2 | 4 |
| | bed | 1 | 89 | 0 | 1 | 0 | 1 | 0 | 4 | 2 | 1 |
| | chair | 0 | 1 | 95 | 0 | 0 | 0 | 1 | 0 | 1 | 1 |
| | desk | 1 | 5 | 2 | 41 | 7 | 3 | 6 | 10 | 23 | 2 |
| Actual | dresser | 1 | 1 | 0 | 2 | 71 | 3 | 16 | 2 | 2 | 1 |
| | monitor | 0 | 1 | 1 | 1 | 2 | 91 | 1 | 1 | 2 | 1 |
| | nightstand | 1 | 1 | 0 | 5 | 20 | 2 | 64 | 1 | 6 | 1 |
| | sofa | 0 | 3 | 1 | 1 | 2 | 3 | 3 | 85 | 0 | 1 |
| | table | 0 | 2 | 1 | 14 | 2 | 0 | 2 | 2 | 77 | 0 |
| | toilet | 0 | 1 | 2 | 1 | 1 | 1 | 1 | 1 | 0 | 92 |

The Table 2 shows the accuracy per category and view. As shown, there is no view with constant low accuracy across all the categories. We did this experiment to check that all the views we selected are actually interesting to perform an accurate classification. Nonetheless, there are isolated combination of views and object that are particularly error prone. This is the case of the view 9 for the bathtub and desk, or the view 19 for the desk.

**Table 2.** Accuracy (%) per class and view of the ModelNet10 test split.

| | | Bathtub | Bed | Chair | Desk | Dresser | Monitor | Night | Sofa | Table | Toilet |
|---|---|---|---|---|---|---|---|---|---|---|---|
| | 1 | 75 | 75 | 95.7 | 48.6 | 64.3 | 97.5 | 68.8 | 62.7 | 78.9 | 91.1 |
| | 2 | 66.7 | 76.5 | 84.2 | 48.1 | 63.5 | 95.9 | 79.4 | 74.5 | 67.4 | 96.1 |
| | 3 | 84.6 | 80 | 91.8 | 68 | 77.3 | 89.1 | 69.6 | 76.6 | 78.7 | 88.9 |
| | 4 | 100 | 90 | 89.1 | 56.5 | 77.8 | 87.7 | 69.6 | 74.5 | 71.7 | 86.4 |
| | 5 | 100 | 95.1 | 83.6 | 54.5 | 57.6 | 87.7 | 55.8 | 69.4 | 69.2 | 88 |
| | 6 | 80 | 92.9 | 83.9 | 50 | 58.5 | 83.3 | 57.5 | 75 | 71.7 | 83.6 |
| | 7 | 81.2 | 95.2 | 96.1 | 50 | 63.5 | 83.3 | 67.4 | 74.5 | 69.2 | 94.8 |
| | 8 | 81.2 | 80 | 92.5 | 54.2 | 75 | 90.7 | 72.5 | 75 | 70.2 | 100 |
| | 9 | 25 | 54.9 | 74.5 | 20 | 57.4 | 69 | 47.1 | 56.9 | 58.8 | 84.3 |
| | 10 | 100 | 87 | 92.5 | 62.5 | 72.2 | 96.2 | 82.9 | 84.4 | 77.3 | 86.4 |
| | 11 | 95 | 93.3 | 100 | 57.9 | 72.5 | 90.6 | 80 | 89.7 | 75.6 | 86.9 |
| | 12 | 90.5 | 86 | 94 | 65.2 | 72.3 | 96.2 | 67.4 | 81.8 | 64.3 | 98.1 |
| View ID | 13 | 65 | 73.5 | 88 | 36.8 | 42.6 | 74.1 | 57.8 | 59.2 | 40 | 83.1 |
| | 14 | 100 | 86 | 98 | 69.6 | 72.3 | 90.9 | 77.1 | 81 | 64.3 | 96.4 |
| | 15 | 100 | 81.5 | 98 | 65.2 | 68.4 | 98 | 70.7 | 92.3 | 69.1 | 94.3 |
| | 16 | 90.9 | 93.8 | 100 | 74.1 | 76.6 | 96.2 | 71.7 | 86 | 75.9 | 98.1 |
| | 17 | 94.4 | 97.7 | 94.2 | 52.8 | 74.5 | 94.1 | 74 | 81 | 64.3 | 96.5 |
| | 18 | 100 | 95.3 | 98 | 57.5 | 69.1 | 91.1 | 73.3 | 83.7 | 76.3 | 100 |
| | 19 | 55.6 | 63.3 | 74.2 | 17.6 | 54.8 | 85.7 | 48.7 | 59.1 | 60 | 83.1 |
| | 20 | 92.9 | 91.3 | 88.7 | 73.9 | 75.5 | 92.6 | 73.9 | 75.6 | 78.4 | 86.7 |
| | 21 | 60 | 75 | 93.6 | 51.4 | 55.6 | 72.5 | 52.9 | 63.5 | 71.2 | 93.1 |
| | 22 | 100 | 91.5 | 98 | 73.1 | 76.5 | 89.5 | 86.4 | 81.4 | 81.2 | 98.2 |
| | 23 | 100 | 95.5 | 96 | 62.1 | 75.5 | 87.9 | 80 | 83.3 | 71.2 | 93.2 |
| | 24 | 50 | 81.8 | 89.6 | 44.1 | 52.5 | 74.5 | 48.8 | 53.5 | 53.1 | 94.5 |
| | 25 | 100 | 88 | 96.1 | 62.5 | 74.5 | 90.9 | 80.5 | 75 | 74 | 96.6 |

We also computed the top-3 accuracy. This metric considers that a sample is correctly classified if the ground truth class if among the three predictions with the higher score. Taking this into account, we

achieved a 94.36% accuracy. This experiment confirms also that, in the cases of bidirectional confusions discussed before, the second or third guess is usually the right one.

The receiver operating characteristic (ROC) curves and area under the curve (AUC) for each category are reported in Figure 7. The ROC curves shows the sensitivity against the 1-specificity, so a perfect classifier would provide the correct class under any detection threshold. As the analysis shows, the classifier is performing nicely on every class, providing 0.8 of true positives rate under a false positive rate of 0.1. Only the category "desk" is not following this trend because, as discussed before, its samples tend to be classified as "table". The area under the curve confirms the accuracy of Par3DNet considering that an AUC of 1 means that the classifier is perfect.

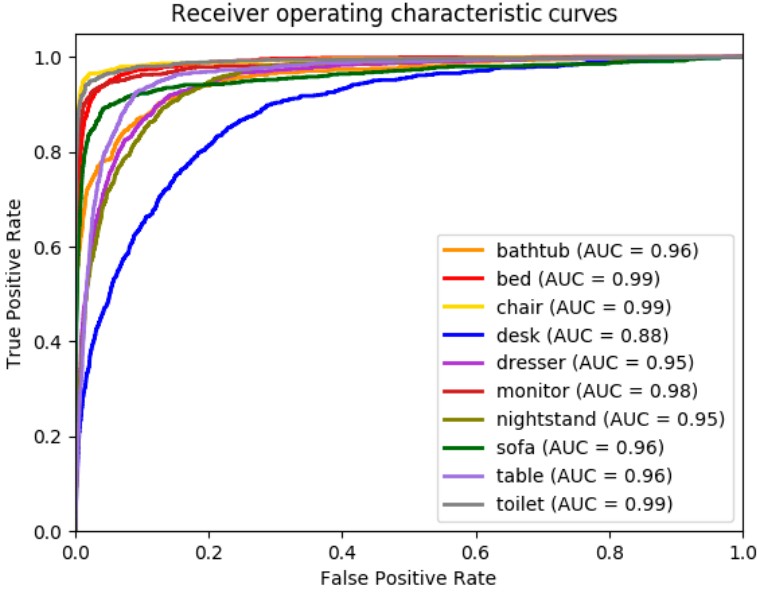

**Figure 7.** Receiver operating characteristic (ROC) curves and area under the curve for each category of the ModelNet10 dataset.

We also compared our system with PointNet [38]. Despite this approach is intended to be fed with complete representations of the objects, we retrained the architecture following the same protocol we used to train Par3DNet. Namely, we retrained the architecture with the partial views of the objects and then tested it. PointNet achieved a top-1 accuracy of 71.25% and a top-3 accuracy of 88.93%, being outperformed by Par3DNet that, as mentioned earlier, achieved 78.39% and 94.36%. It is also worth noting that the number of input points for PointNet is fixed, so it cannot be fed with different resolution point clouds. However, Par3DNet can be fed with point clouds composed of an arbitrary number of points.

It should be noted that our method takes as an input a partial view of the object. This fact makes the comparison unfair with the participants of the ModelNet10 challenge as they consider the full object. So, to enable an easy and fair comparison with the approaches submitted to the challenge, we adapted our method to work on full objects. To do so, we computed the accuracy per model by averaging the predictions for each view, and reporting the category with the highest score. We achieved a 88.98% accuracy following this methodology. It is important to remark that our method is not the best performer as for the challenge means, but our goal is to provide a method that actually works on real life data. We only performed this experiment in the sake of comparison. An updated list of the methods submitted to this challenge could be seen in [17].

### 4.2. Results on Noisy ModelNet10 Dataset

The point clouds returned by the method explained in Section 3.2, makes the tridimensional representations too perfect. This is understandable as the models are synthetic. Nonetheless, the data

provided by actual sensors yield some noise. To benchmark the accuracy of our proposal in presence of noise, we artificially disturbed the point clouds at different levels. To do so, we artificially added Gaussian noise. The mean is the position of each point and the variance was modified for each experiment. Resultant samples for different noise levels can be seen in Figure 8. It is worth noting that the noise levels are arbitrary values as the point clouds come from synthetic models and do not have an actual metric scale. Nonetheless, the samples were resized so they all share the same size and are affected in the same way for each noise level.

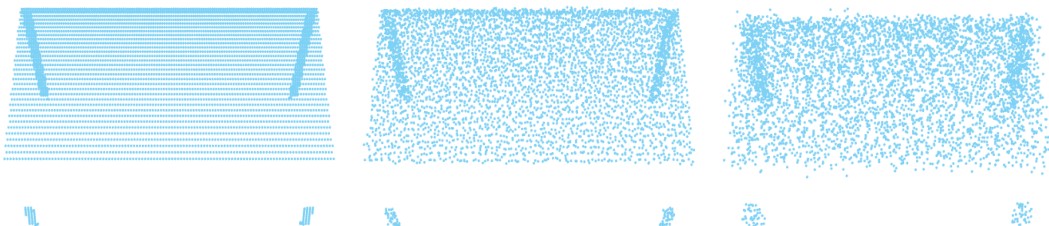

**Figure 8.** The leftmost image is a random sample with no added noise, as captured by the virtual 3D camera. The image in the center corresponds to a noise level of 0.25. Finally, the leftmost image is disturbed by a factor of 1.

The results are as shown in Table 3. The accuracy in this experiment decreases as the noise level is augmented. This is expected as if the noise is too high, the geometric features of the objects are lost and the object becomes unrecognizable. Nonetheless, our method is able to deal with some level of noise until 0.30, where the accuracy drastically decreased. The noise tolerance is provided by the binary voxelized representation. As long as the noisy points are near the optimal position, it will be in the same voxel or in the adjacent ones. If the noise is too high, the points would be very far from the optimal location. It is important to remark that the model was not retrained with noisy samples, so the noise tolerance is provided by our method and not by the data.

**Table 3.** Accuracy obtained for the ModelNet10 test split after introducing different levels of noise for Par3DNet and PointNet.

| Noise Level | 0.01 | 0.05 | 0.15 | 0.20 | 0.25 | 0.30 | 0.40 | 0.50 | 0.75 | 1.00 |
|---|---|---|---|---|---|---|---|---|---|---|
| Par3DNet Accuracy (%) | 78.41 | 78.14 | 76.27 | 73.76 | 70.9 | 56.79 | 54.81 | 52.35 | 45.63 | 43.61 |
| PointNet Accuracy (%) | 56.27 | 56.36 | 56.10 | 56.15 | 56.20 | 56.13 | 55.86 | 55.97 | 55.19 | 54.50 |

Table 3 shows the performance achieved by PointNet too. In this case, PointNet suffers an accuracy drop of around a 15% just by adding a slight noise level of 0.01 (compared with the performance provided by the same architecture in presence of no noise at all). The accuracy is more or less kept along the range of tested noise levels. It is worth noting that, as the authors of PointNet suggest, a live data augmentation method is carried out by randomly jittering the points. Thus, in this case, the eventual robustness to noise is provided by the data and not by the method itself.

### 4.3. Results on the ObjectNet Dataset

In order to test the generalization capabilities of our approach, we tested it with the ObjectNet [36] dataset. This is a dataset for 3D object recognition and is composed of 100 categories, 90,127 images, 201,888 instances of objects in these images and 44,147 3D shapes. Objects in the images in this database are aligned with the 3D shapes, and the alignment provides both accurate 3D pose annotation and the closest 3D shape annotation for each 2D object. Nonetheless, we only are interested in the 3D objects. The authors provided not only full models of the objects but the parts of it too. To serve our purpose, we took the full models of the following categories: bathtub, bed, chair, sofa, table and toilet. The rest

of the categories are omitted as they are not considered by our approach. Finally, only 65 models were used for this experiment.

We preprocessed the CAD models as explained in Section 3.2 and fed the resultant point clouds to our architecture. The accuracy was 64.4% and the results are as shown in Table 4.

**Table 4.** Confusion matrix of the results achieved by our model when fed the ObjectNet dataset.

|  |  | **Predicted** |  |  |  |  |  |  |  |  |  |
|---|---|---|---|---|---|---|---|---|---|---|---|
|  |  | **Bathtub** | **Bed** | **Chair** | **Sofa** | **Table** | **Toilet** | **Desk** | **Dresser** | **Monitor** | **Nightstand** |
| **Actual** | bathtub | 32 | 22 | 2 | 16 | 4 | 5 | 2 | 4 | 9 | 4 |
|  | bed | 2 | 76 | 1 | 0 | 1 | 5 | 0 | 3 | 2 | 10 |
|  | chair | 0 | 0 | 100 | 0 | 0 | 0 | 0 | 0 | 0 | 0 |
|  | sofa | 0 | 2 | 8 | 48 | 0 | 14 | 0 | 0 | 24 | 4 |
|  | table | 0 | 0 | 0 | 0 | 96 | 0 | 2 | 0 | 0 | 2 |
|  | toilet | 8 | 3 | 0 | 0 | 5 | 78 | 0 | 1 | 0 | 5 |

The accuracy of 32% for the bathtub category is due to the inclusion of different objects that are actually not bathtubs. For instance, jacuzzi tubs and swimming pools were labeled as bathtubs in this dataset. In addition to this, this is the category with the lowest performance also in the results for the ModelNet10 experiment. The sofa category also achieved a low accuracy for the same reason. There are objects like armchairs and other objects labeled as sofa that should not be included in that category. Some of these mislabeled objects are shown in Figure 9. Overall, we also found that the CAD models of this dataset lack of detail and are of poor quality compared with ModelNet. The accuracy of the rest of the categories is high.

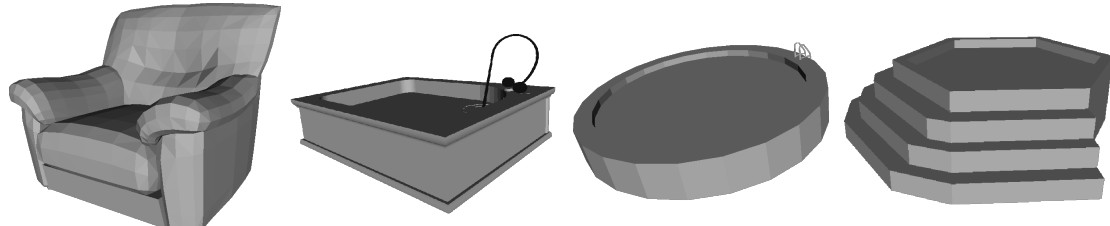

**Figure 9.** Some samples of the ObjectNet dataset that are not correctly labeled. The first sample was labeled as sofa and the rest as bathtub.

### 4.4. Results on the ShapeNetCoreV2 Dataset

We also tested our approach with the ShapeNetCore [39] dataset to further benchmark the generalization capabilities of our proposal. ShapeNetCore is a subset of the full ShapeNet dataset with single clean 3D models and manually verified category and alignment annotations. It covers 55 common object categories with about 51,300 unique 3D models. As in the case described in Section 4.3, we only took the categories considered by our approach, namely, the categories bathtub, bed, chair, table and sofa. In this experiment, there were involved more than 19,400 samples that were converted to point clouds following the procedure described in Section 3.2.

The accuracy our approach got with this dataset was 71.80%. However, as show in Table 5, the bathtub yielded a similar result than in the rest of the experiments. In addition to being the category with the fewest number of samples in the training set, in this case, the bathtub subset also included a large quantity of swimming pools, jacuzzis, bathroom furniture and other objects that are not exactly bathtubs. In fact, the *synset* for this category is "bathtub, bathing tub, bath, tub". A similar case can be seen for the bed category. In this subset, half of the samples were bunk beds, which were not considered by our model. Some of these samples that are not compatible with our labeling can be

seen in Figure 10. In this case it can also be seen a confusion between table and desk. Nonetheless, the rest of the categories achieved a high accuracy. The top-3 accuracy was 87.43%.

**Table 5.** Confusion matrix of the results achieved by our model when fed the ShapeNetCore dataset.

| | | Predicted | | | | | | | | | |
|---|---|---|---|---|---|---|---|---|---|---|---|
| | | **Bathtub** | **Bed** | **Chair** | **Sofa** | **Table** | **Toilet** | **Desk** | **Dresser** | **Monitor** | **Nightstand** |
| **Actual** | bathtub | 52 | 7 | 4 | 11 | 4 | 5 | 3 | 7 | 4 | 6 |
| | bed | 1 | 46 | 8 | 6 | 11 | 4 | 9 | 3 | 4 | 9 |
| | chair | 1 | 4 | 78 | 2 | 2 | 6 | 1 | 1 | 3 | 3 |
| | sofa | 2 | 5 | 1 | 82 | 1 | 1 | 2 | 2 | 3 | 1 |
| | table | 1 | 4 | 2 | 3 | 67 | 1 | 12 | 3 | 2 | 6 |

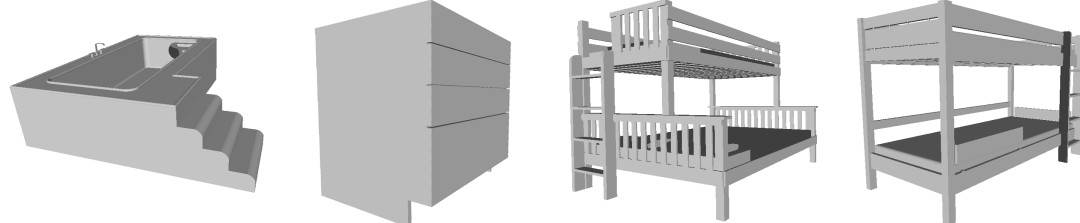

**Figure 10.** Some samples of the ShapeNetCore dataset that are not accurately labeled. The first and second sample were labeled as bathtub and the rest as bed.

### 4.5. Results on Real Data

Finally, we also put to test our approach with real data. In this preliminary experiment, we used a Kinect camera to capture point clouds of several objects from different points of view. Then, we manually segmented them so the background and undesirable artifacts are removed. The resultant point clouds only depict the partial view of the objects of interest, as shown in Figure 11. These point clouds were then converted to the binary voxel representation and fed to the architecture.

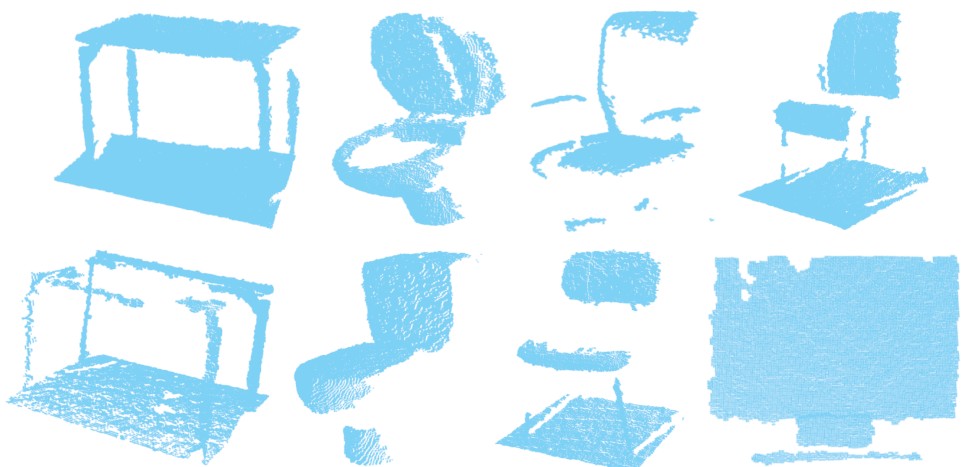

**Figure 11.** Some random samples provided by a Kinect sensor we captured to test our approach.

As shown in Figure 11, the samples are highly cluttered and yield severe noise issues and self-occlusions. Despite of that, our approach was able to correctly classify the 72.72% of the real samples. It is worth noting that the model was trained on synthetic data and is tested with real data.

## 5. Conclusions and Future Work

In this work we introduce Par3DNet, a 3DCNN for object recognition that take as an input pure tridimensional data. Our strongest contribution is that our architecture is able to classify partial views of the object, as if the input data were captured by a real sensor. Actually, preliminary tests indicate that our model performs accurately on point clouds provided by a Kinect camera. We focused on the ModelNet10 as this is the most used state of the art dataset for 3D object recognition. We also successfully tested our approach with ObjectNet and ShapeNetCore datasets.

Despite our model performing accurately overall, it has some limitations. For instance, the bathtub category does not perform as good as the rest of the categories. However, this is understandable as this category has the lowest number of samples by a large margin. There also is a consistent confusion between table and desk. As discussed before, these two categories represent barely the same object. We considered them as two different categories to follow the ModelNet10 rules and to allow an easy comparison with other algorithms. The source code of the approach and the model we used for all the experiments can be downloaded from our public repository https://bitbucket.org/fgd5/par3dnet.

Regarding the computation time, a feedforward step of the architecture only takes 9 ms in the hardware described before. Nonetheless, the generation of the binary voxel representation takes 140 ms. Finally, the full pipeline takes 170 ms from end to end.

As for the future work, we plan to involve noisy data at training time to further improve the accuracy and robustness of the system. We also plan to include more categories and samples by merging different datasets. In addition, we are also after gathering, and potentially recording, a proper dataset of segmented 3D objects provided by real sensors. We will also append a pose estimation system to the network, so that the architecture is able to provide not only the classification of the input point cloud, but the position and orientation of the objects in the camera's coordinate frame too. Finally, we also plan to develop a method to automatically segment the object from the background, as this is being done manually so far.

**Author Contributions:** Conceptualization, methodology, writing—review and editing, supervision, project administration and funding acquisition, M.C.; Software, validation, formal analysis, resources, visualization, writing—original draft preparation, F.E.; Conceptualization, writing—original draft preparation, software, validation, formal analysis, data curation, visualization, F.G.-D. All authors have read and agreed to the published version of the manuscript.

**Funding:** This work has been funded by the Spanish Government TIN2016-76515-R grant for the COMBAHO project, supported with Feder funds. It has also been supported by Spanish grants for PhD studies ACIF/2017/243 and FPU16/00887.

**Acknowledgments:** The authors would like to thank NVIDIA (Santa Clara, California, USA) for the generous donation of a Titan Xp and a Quadro P6000.

**Conflicts of Interest:** The authors declare no conflict of interest.

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
