# Peer review of "Par3DNet: Using 3DCNNs for Object Recognition on Tridimensional Partial Views"

_applsci, doi:10.3390/app10103409_

Round 1

Reviewer 1 Report

The authors introduced a method for automatic 3D object detection. Overall, this topic is important and obtained increased attention. However, a few sections where critical details have been left out, which should be included before the paper is published.

The major concerns are shown below:

  1. Line 126, the explanation that the authors selected ModelNet10 dataset for testing is not powerless. There are several these kinds of datasets such as:
    1. SUNRGBD: http://rgbd.cs.princeton.edu/
    2. SCANNET: http://www.scan-net.org/
    3. S3DIS dataset: http://buildingparser.stanford.edu/dataset.html

These datasets were wildly used to evaluate the performance of the proposed 3D structure.

  1. In this paper, the authors did not provide comparison between this method and other popular used structures such as PointNet++, VoteNet, and PVCNN.

Zhou, Y., Tuzel, O., 2017. VoxelNet: End-to-End Learning for Point Cloud Based 3D Object Detection, IEEE/CVF Conference on Computer Vision and Pattern Recognition. IEEE, Salt Lake City, UT, USA pp. 1-10.

Qi, C.R., Litany, O., He, K., Guibas, L.J., 2019. Deep Hough Voting for 3D Object Detection in Point Clouds. arXiv preprint arXiv:1904.09664.

Liu, Z., Tang, H., Lin, Y., Han, S., 2019. Point-Voxel CNN for efficient 3D deep learning, Advances in Neural Information Processing Systems, pp. 963-973.

These are the major limitations of this paper!

  1. Introduction is too simple; No references are cited! In Related works, many important previous studies are not reviewed.

Author Response

Please, see attachment

Reviewer 2 Report

The proposed method takes as an input a 3D volume and trains them with 3D Convolutional Neural Networks.
The proposed approach is quite interesting to the reviewer. It is recommended to clarify the manuscript as follows:

1) It is recommended to add some references in the introduction.
2) A more detail description is required in Fig. 1.
3) It is better to add one more paragraph to explain the motivation and the uniqueness of the proposed approach at the end of Section 2.
4) It is recommended to justify a dropout ratio with 0.2.
5) how did you get the 25 most relevant views? What is the criterion to get these views, e.g. based on 360 angles or something else? It is recommended to clarify the relevant views.

The authors showed lots of experimental results to demonstrate the performance of the proposed methods.
It is recommended to show ROC curves if applicable.
